# Dysbiosis of the Fecal and Biliary Microbiota in Biliary Tract Cancer

**DOI:** 10.3390/cancers14215379

**Published:** 2022-10-31

**Authors:** Zensho Ito, Shigeo Koido, Kumiko Kato, Toshitaka Odamaki, Sankichi Horiuchi, Takafumi Akasu, Masayuki Saruta, Taigo Hata, Yu Kumagai, Shuichi Fujioka, Takeyuki Misawa, Jin-zhong Xiao, Nobuhiro Sato, Toshifumi Ohkusa

**Affiliations:** 1Department of Internal Medicine, Division of Gastroenterology and Hepatology, The Jikei University School of Medicine, Kashiwa Hospital, Kashiwa City 277-8567, Chiba, Japan; 2Institute of Clinical Medicine and Research, The Jikei University School of Medicine, Kashiwa City 277-8567, Chiba, Japan; 3Gut Microbiota Department, Next Generation Science Institute, Morinaga Milk Industry Co., Ltd., Higashihara 252-8583, Zama, Japan; 4Department of Internal Medicine, Division of Gastroenterology and Hepatology, The Jikei University School of Medicine, Tokyo 105-8471, Japan; 5Department of Surgery, The Jikei University School of Medicine, Kashiwa Hospital, Kashiwa City 277-8567, Chiba, Japan; 6Department of Microbiota Research, Juntendo University Graduate School of Medicine, 3-3-1 Hongo, Tokyo 113-0033, Japan

**Keywords:** biliary tract cancer, fecal microbiota, bile microbiota, *Enterobacteriaceae*, polyketide synthase (pks), carcinogenic bacteria

## Abstract

**Simple Summary:**

There is no reliable data on the dysbiosis of fecal microbiota in biliary tract cancer. We present a metagenomic study to simultaneously analyze the microbiota in bile and feces and found that patients with biliary tract cancer had more *Enterobacteriaceae* and less *Clostridia*, including butyrate-producing bacteria such as *Faecalibacterium* and *Coprococcus*. Furthermore, metagenomic analysis revealed that the strains isolated from bile harbored genes encoding carcinogenic bacterial colipolyketide synthases (pks). The biliary microbiota is heavily influenced by the colonic flora, and carcinogenic bacteria may be a new risk factor for biliary tract cancer.

**Abstract:**

Characteristic bile duct and gut microbiota have been identified in patients with chronic biliary tract disease. This study aimed to characterize the fecal and bile microbiota in biliary tract cancer (BTC) patients and their relationship. Patients with BTC (*n* = 30) and benign biliary disease (BBD) without cholangitis (*n* = 11) were included. Ten healthy, age-matched subjects were also recruited for fecal microbiota comparison. The fecal and bile duct microbiotas were analyzed by sequencing the 16S rRNA gene V3-V4 region. Live bacteria were obtained in the bile from three BTC patients by culture, and metagenomics-based identification was performed. Linear discriminant analysis effect size showed a higher *Enterobacteriaceae* abundance and a lower *Clostridia* abundance, including that of *Faecalibacterium* and *Coprococcus*, in the BTC patients than in the other subjects. Ten of 17 operational taxonomic units (OTUs) assigned to *Enterobacteriaceae* in the bile were matched with the OTUs found in the BTC subject fecal samples. Furthermore, a bile-isolated strain possessed the carcinogenic bacterial colipolyketide synthase-encoding gene. *Enterobacteriaceae* was enriched in the BTC feces, and more than half of *Enterobacteriaceae* in the bile matched that in the feces at the OTU level. Our data suggests that fecal microbiota dysbiosis may contribute to BTC onset.

## 1. Introduction

Biliary tract cancer (BTC) is a cancer with a poor prognosis. It is often unresectable at diagnosis and has a 5-year overall survival rate of 10% or less [1,2]. The incidence of BTC is considered high, 3 per 100,000 in Hispanic and Asian populations [3,4], but this incidence is increasing in not only Asia but also Western European countries [1,5,6]. BTC has a high potential for metastasis and invasion, and because of its anatomic location and spread along the bile ducts, it is difficult to resect completely by surgery. The standard practice for advanced BTC is cisplatin or gemcitabine, but the response to these chemotherapies is poor, resulting in a 5-year survival rate of just under 10% [5]. Several risk factors for BTC are known. Diseases that increase the risk for BTC include cirrhosis, congenital liver fibrosis, metabolic disease, and liver diseases such as primary sclerosing cholangitis (PSC) [2,7,8]. Genetic studies have shown that pathological genetic mutations such as mutations in BRCA1/2, MLH1, MSH2, and TP53 are found in BTC [9], but genetic factors alone cannot explain the onset of this disease, and the etiology of BTC remains unknown.

Recent studies using next-generation sequencing (NGS) have shown that the gut microbiota of patients with liver disease is not only altered at both the upper gastrointestinal and bile duct levels but also promotes changes in the microbiota composition of the colon [10]. In addition, although bile has been widely regarded as sterile, studies in patients with PSC have shown an association between bile acids and bile bacteria [11]. Several bile studies have shown that Proteobacteria, Firmicutes, and *Bacteroides* are the major constituents in bile [12]. In a study comparing patients with intrahepatic BTC and patients with cirrhosis, two genera, *Lactobacillus* and *Alloscardovia*, were reported as potential prognostic markers [13]. These findings indicate that the bile microbiota is related to hepatobiliary diseases, including BTC. However, the distribution of biliary microbiota in BTC and its comparison with fecal microbiota have not been studied. Feces comprises one of the specimens that can be easily evaluated. It is therefore essential to identify the components of the bile and fecal microbiota and the specific bacteria associated with BTC for prevention and therapeutic development. The purpose of this study was to investigate the fecal and bile microbiota in selected cohorts of patients experiencing intrahepatic cholangiocarcinoma, extrahepatic cholangiocarcinoma, and gallbladder cancer as BTC and cholecystectomy for benign biliary tract disease (BBD). In addition, carcinogenic bacteria in the bile of patients with BTC were studied.

## 2. Materials and Methods

### 2.1. Patients

Fifty-four consecutive patients were seen for biliary tract diseases. Of these, 13 patients met the following exclusion criteria and were excluded from the analysis. Finally, 30 patients with BTC and 11 patients with BBD were included. Ten healthy, age-matched subjects were also recruited for fecal microbiota comparison (Figure 1).

All patients with biliary tract disease and controls were recruited at Jikei University Kashiwa Hospital. The diagnosis of BTC was based on cholangiography, the presence of typical cholangial lesions on tissue biopsy, contrast CT findings, and elevated tumor markers. The exclusion criteria was a patient age of less than 18 years, acute bacterial cholangitis, severe medical comorbidities, and previous receipt of treatment interventions, such as endoscopic retrograde cholangiography (ERC) and surgical or anticancer therapy. Patients could not receive antibiotics within two months prior to participation in the study. This interval was determined to be a sufficient period of time for the gut microbiota to recover from the effects of antibiotic administration.

In the BTC group of this study, there were four patients whose chief complaint was jaundice. This is because the majority of patients were suspected of having BTC based on physical examinations and imaging tests. In BTC patients with jaundice who underwent ERC before surgery, bile samples were obtained at the time of initial ERC. Indications for ERC in patients with BTC included the purpose of bile stasis treatment or BTC diagnosis; during ERC, bile aspiration was performed without the use of contrast or interventional antibiotic prophylaxis. No patients with BTC were sampled for acute suppurative cholangitis. The BBD group was established as controls and included those with surgical intent for gallbladder stones or gallbladder polyps. The BBD group also enrolled patients who visited Jikei University Kashiwa Hospital. The BBD group included patients diagnosed with gallbladder/common bile ductal stones or gallbladder polyps without cholangitis conditions requiring emergency treatment; the BBD group included patients who had not been administered preoperative antibiotics for 2 months. Thus, no preoperative antibiotics were administered. This is because we also believe that antibiotic administration may affect bile and stool cultures. Patients with chronic cholecystitis or acute cholecystitis, a condition that produces persistent inflammation, were excluded. The healthy subject group was also enrolled from Jikei University Kashiwa Hospital. They had no history of abdominal surgery, severe medical complications, endoscopic retrograde cholangiography (ERC), surgical treatment, or anticancer therapy, taking into account changes in intestinal bacteria. In addition, the healthy subject group did not receive antibiotics more than 2 months prior to participation in the study. This criterion was the same as the BTC and BBD group criteria. Stool samples were collected on the morning of the study day, and the method of stool collection was the same for the BTC, BBD, and healthy subject groups. All subjects provided written informed consent, and the study was approved by the clinical research ethics committee of the Jikei University School of Medicine and Kashiwa Hospital, the Jikei University School of Medicine (number 29-146 (8762)). This study was conducted in accordance with the Declaration of Helsinki.

### 2.2. Analysis of Fecal Microbiota

Fecal samples were collected on the morning of the hospital visit, and a stool sample aliquot was mixed with 1 mL of guanidine thiocyanate (GuSCN) solution (TechnoSuruga Laboratory Co., Ltd., Shizuoka, Japan), immediately frozen at −80 °C, and stored until analysis. DNA extraction from the human fecal samples was performed using the bead-beating method as previously described, with some modifications [14]. Briefly, 150 μL of fecal sample in GuSCN solution was vigorously vortexed with 300 mg of glass beads (AS ONE BZ-01) and 500 μL of Tris-EDTA (TE, pH 9.0) buffer-saturated phenol (Fujifilm, Wako Pure Chemicals) using a FastPrep-24 (Funakoshi Corporation) for 30 s at power level 5. After centrifugation at 10,000× *g* for 10 min, 400 μL of the supernatant was extracted with 500 μL of phenol–chloroform, and 250 μL of the supernatant was precipitated with isopropanol. The purified DNA was suspended in 100 μL of TE buffer (pH 8.0).

### 2.3. Bile Collection Procedure and Biological Sample Acquisition

Bile was collected during endoscopic or surgical treatment. Bile samples were collected in Techno Suruga Lab containers as in the fecal microbiota analysis. Some bile was also cultured simultaneously with the culture method using the medium described below. All endoscopic surgeries were performed under conscious sedation. Endoscopic retrograde cholangiography was performed with a standard video duodenoscope (TFJ 260-V, Olympus, Tokyo, Japan). A Cook cannula (Cook, Washington, DC, USA) and a Boston guidewire (Boston Scientific, Tokyo, Japan) were used for selective cannulation of the bile duct. A bile sample was aspirated prior to the application of the contrast agent. Antibiotic prophylaxis was applied intravenously after bile samples were obtained and endoscopic scrutiny and treatment were completed. All cholecystectomies or BTC surgeries were performed under general anesthesia, either open or laparoscopic. Bile samples were aspirated and collected by aseptic manipulation. The biological samples were stored immediately after sampling at −80 °C until DNA extraction. ERC was not performed in all patients. Therefore, some patients in the BBD and BTC groups had bile collected at the time of surgery (Figure 1).

### 2.4. Bile Culture Assay

Bile was collected intraoperatively by sterilization from consenting patients with BTC or BBD who underwent surgery. A portion of the bile was cultured immediately. For anaerobic culture, bile was collected in Kenky Porter II (KP-C0402, Terumo Co., Ltd.), inoculated with 100 μL of Kenky Porter II in the following medium, and incubated anaerobically for 48 h. The resulting colonies were collected, and DNA was extracted. Sheep blood agar medium (E-MP35, Eiken Chemical), GAM agar medium (05420, Nissui), gentamicin (20 μg/mL) (Sigma, G1272)-supplemented GAM agar medium, and FM agar medium (05441, Nissui) were used. For aerobic cultures, bile was collected in a stool collection container (Technosulga Lab.) for intestinal microbiota testing, inoculated with 100 μL of sheep blood agar medium (Eiken Chemical Co., Ltd.) and BTB agar medium (E-MA84, Eiken Chemical Co., Ltd.), and incubated aerobically for 24 h. DNA was extracted from the mixed colonies in each sample.

### 2.5. Microbiota Analysis

Amplicon sequencing of the V3-V4 regions of the bacterial 16S rRNA gene was performed with an Illumina MiSeq instrument, as described previously [14]. Data were analyzed in the QIIME2 software package [15] (ver. 2017.10). The reads were mapped to the PhiX 174 sequence and the Genome Reference Consortium human build 38 (GRCh38) by the Bowtie-2 program [16] (ver. 2–2.2.4), and potential chimeric sequences were removed from acquiring the Illumina paired-end reads by using DADA2 [17]. Thereafter, 30 and 90 bases of the 3′ region of the forward and reverse reads were trimmed, respectively. Taxonomic classification was performed using a naive Bayes classifier trained on Greengenes 13.8 [18] with a 99% threshold for operational taxonomic unit (OTU) full-length sequences. An estimation of alpha diversity and a principal coordinate analysis (PCoA) for beta diversity were also performed using QIIME2.

### 2.6. Detection of Polyketide Synthase (pks) Genomic Islands in Cultured Bacteria Isolated from Bile Acid

The library construction for an Illumina MiSeq instrument and subsequent de novo assembly of raw reads by the CLC Genomics Workbench (v 8.0) software package (Qiagen, Valencia, CA, USA) were performed as previously described [19]. The open reading frame (ORF) prediction and annotation were performed using the DDBJ Fast Annotation and Submission Tool (DFAST) with the default settings [20]. Colibactin genomic islands were detected by BLASTP analysis against NCBI reference sequences WP_001217110.1, WP_000357141.1, WP_001518711.1, WP_001297908.1, WP_000982270.1, WP_001297917.1, WP_000337350.1, WP_000159201.1, WP_001304254.1, WP_000829570.1, WP_001468003.1, WP_000222467.1, WP_001297937.1, WP_000217768.1, WP_001327259.1, WP_001029878.1, WP_002430641.1, and WP_000065646.1. PCR for corroboration of the existence of *pks* islands was performed as previously described [21]. The region of primers for the *pks* island is shown in Appendix A.

### 2.7. Statistical Analysis

A permutational multivariate analysis of variance (PERMANOVA) based on the UniFrac distances was used to evaluate interindividual variability among groups by using the ‘adonis’ function in the vegan R package (ver. 3.3.0), and *p* values of <0.05 were considered statistically significant. A linear discriminant analysis (LDA) effect size (LEfSe) was performed with default parameters to identify microbial taxa that were differentially abundant among groups [22]. An LDA score >2.0 and false discovery rate (FDR)-corrected *p* value < 0.05 were considered to indicate significance.

### 2.8. Data Deposition

DNA sequences corresponding to the 16S rRNA gene and metagenome data have been deposited in DDBJ under accession numbers DRA011518 and DRA011520, respectively.

## 3. Results

### 3.1. Subject Background

Table 1 shows the subject background in addition to blood parameters. There was no significant difference in age (*p* = 0.054), sex (*p* = 0.093), or body mass index (BMI) (*p* = 0.061) among the three groups. Some blood parameters were different (Table 1). Cholangitis is a common complication of BTC, and its coexistence is an important concern in the evaluation of the bile microbiota. There was no difference in white blood cell (WBC) levels as an indicator of the infection status in the BTC group. The decrease in hemoglobin and albumin levels and the increase in C-reactive protein (CRP) levels observed in the BTC group may indicate carcinoma status.

### 3.2. Fecal Microbiota

We first examined the composition of the fecal microbiota in the BTC, BBD, and healthy groups. Both weighted and unweighted UniFrac PCoA showed a significant difference in fecal microbiota composition (Figure 2).

LEfSe indicated a higher abundance of *Gammaproteobacteria*, including *Enterobacteriaceae*, and a lower abundance of *Clostridia*, mainly composed of *Lachnospiraceae*, in the BTC group than in the other groups (Figure 3A−D).

Notably, some butyrate-producing bacteria, such as *Faecalibacterium* and *Coprococcus*, were enriched in the healthy group. No apparent difference was observed in the alpha diversities based on the Shannon, observed OTUs, Chao1, and faith_pd indices (Appendix A).

### 3.3. Altered Biliary Microbiota in Patients with BTC

We then conducted biliary microbiota analysis in 8 patients in the BTC group and 10 patients in the BBD group. Of these, we succeeded in the amplification of 16S rRNA genes in four of the eight patients with BTC and in three of the ten patients with BBD, indicating bacteria in some of the bile ducts. The proportion of bacterial species in the bile microbiota varied between individuals (Appendix A), but an OTU assigned to *Enterobacteriaceae* was enriched in the bile samples of the BTC group compared to the BBD group (Figure 3E). To predict the source of the bacteria in the bile duct, we subsequently compared the bile and fecal microbiota. PCoA showed an obvious difference between them (Appendix A); however, there were some common OTUs assigned to *Enterobacteriaceae* in the fecal and bile microbiota (Table 2). Notably, the *Enterobacteriaceae* OTUs were matched in 1 of the 11 (9.1%) samples in the BBD group and 10 of the 19 (52.6%) samples in the BTC group. The result of comparative analysis of the fecal and biliary microbiota is reference data due to the small number of cases. Additional experiments cannot be performed in this study. However, we believe that increasing the number of cases and evaluating the results will be a future issue.

### 3.4. Detection of a Pks Genomic Island in an Isolate from Bile Acid

To prove the existence of live bacteria in the bile of the patients, we subsequently tried to isolate bacteria from the bile samples. Colonies were detected from the bile samples of subjects 37, 40, and 41. Finally, we performed a metagenomics analysis to reveal the features of the isolates from bile acids. Our sequencing effort demonstrated a base sequence-coded *pks* genomic island, which was responsible for colibactin production in mixed isolates from the bile acid of subject 41, although the detected *pks* island was separated on the two contigs (contigs 163 and 33, Appendix A). To corroborate the missing area (part of *clbL* and *clbK*) in the metagenomic data, we conducted additional PCRs using six primer pairs for the whole region of the *pks*-island, as shown in Appendix A. All of the PCR results were positive, suggesting the existence of the whole *pks* genomic island in the isolate from bile acid. No positive findings were observed in the other bile acid samples.

## 4. Discussion

It has long been suggested that cholecystitis and cholangitis are caused by an intestinal bacterial infection. Therefore, we believe that BTC should be investigated in relation to the gastrointestinal microbiota, including the fecal intestinal microbiota and the bile microbiota. In addition, BTC is often accompanied by chronic inflammation of the gallbladder and bile ducts, and chronic inflammation is reported to be caused by infections of intestinal bacteria, including *Escherichia coli* [2]. In other words, cholecystitis and cholangitis are risk factors for BTC, and intestinal bacteria are the main cause of this risk.

BTC is a disease that occurs in the bile ducts, which have a large mucosal barrier. Diseases that cause chronic inflammation in the bile ducts are known to result in an altered microbiome in the bile ducts. This phenomenon may provide an argument explaining why cholangitis is one of the risk factors for BTC and may contribute significantly to the nongenetic risk associated with BTC. Previous studies of the microbiome of patients with BTC have reported an increased abundance of *Lactobacillus*, *Actinomyces*, and others in their gut microbiome [13]. Regarding the bile microbiota, the relative proportion of *Fusobacteria*, *Acidobacteria, Planctomycetes*, etc., was reported to be increased in patients with BTC [12]. Previous studies have demonstrated the association between abnormalities of the biliary microbiota and BBD [23], validating the possibility that abnormalities of the biliary microbiota are the main cause of the presence of biliary diseases. However, a common view on whether changes in the fecal microbiota are signs of abnormalities in the bile duct microbiota or whether changes in the fecal microbiota affect the composition of the bile duct microbiota has not been established.

These reports did not study the bacterial microbiota simultaneously with feces and bile. Because feces and bile mutually interact, we felt that it was necessary to study both microbiota constituents simultaneously to study the microbiota constituents responsible for BTC. This is the first study to simultaneously examine the fecal and bile microbiota in patients with BTC and to more clearly evaluate the microbiota of patients with BTC by performing bile cultures. A variety of factors affect the gut microbiota [24,25,26]. Probiotics and antibiotics are also involved in altering the composition and/or metabolites of the gut microbiota [27,28]. With regard to the history of probiotic and antibiotic use, we included individuals who had not received any probiotics or antibiotics in the 2 months prior to sample collection; the 2-month period was established to allow for recovery from the effects of antibiotics on the intestinal microbiota [29,30].

In this study, the microbiota was analyzed among three groups: patients with BTC, patients with BBD, and healthy individuals. Age and sex, which are factors affecting the gut microbiota, were not biased among the three groups. The fact that the background factors were well matched among the three groups allowed us to accurately evaluate the bacterial microbiota in the disease and to analyze it precisely. Since this study compared bacterial microbiota among three groups, we believe that a statistical analysis of three-group comparisons is appropriate. *ANOVA* was used for the three-group comparison of age. For the validation of the bile microbiota, we compared the bile microbiota between the BTC and BBD groups. The reason for the exclusion of healthy subjects was that it is unethical to insert a biliary endoscope into a healthy person for the sole purpose of bile collection, which would be highly invasive. The compositions of the fecal microbiota were significantly different (Figure 2), suggesting that the development of BTC, i.e., harboring cancer, may lead to changes in the bacterial microbiota at various sites. Therefore, it is worthwhile to examine the fecal microbiota in detail in patients with BTC. The cladogram analysis up to the genus level produced by LEfSe for the patients with BTC, patients with BBD, and healthy individuals did not reveal any significant species in the BBD group alone. *Enterobacteriaceae* was predominantly detected in the BTC group, while *Lachnospira*, *Faecalibacterium*, and *Coprococcus* were significantly enriched in the healthy group (Figure 3). The bile microbiota was also the focus point in the present study. Because bile salts are toxic to bacteria, an equilibrium of modified bile salts is reached that allows commensal bacteria to survive but rebuffs invading pathogens [31]. Therefore, the number of bacteria in bile is thought to be very small compared to the number of bacteria in feces. This is one of the reasons why the analysis of bile bacteria is difficult. It was not possible to evaluate which method was more likely to amplify 16S rRNA, ERC or surgical collection, due to the small number of cases in this study. Moreover, there were no reports that have examined this possibility. Surprisingly, we detected the 16S rRNA gene in 7 of 18 bile samples from biliary tract disease patients. These results were similar to those reported in Japanese patients with PSC, where bacterial PCR was positive in less than 60% of cases [32]. The culture method corroborated the existence of live *Escherichia coli* in three bile samples, although the effect of bile duct organ characteristics on the defense against infection, i.e., bacterial growth in the presence of bile salts, is notable [33]. This result is consistent with the clinical prevalence of *Escherichia coli*, *Klebsiella* spp., *Enterobacter* spp., *Enterococcus* spp., and *Streptococcus* spp. and other gram-negative bacteria as the causative agents of cholangitis [34]. This could provide an argument for the influence of the gut microbiota on the bile duct microbiota. Similar to the gut-brain axis [35], the gut-bile duct axis might exist. An increased risk of the late onset of BTC after papillary sphincteroplasty and common bile duct duodenal anastomosis has been reported, which could provide an argument for the influence of changes in the bile duct microbiota on BTC [36,37].

Interestingly, 10 of 19 OTUs (53%) assigned to *Enterobacteriaceae* in the bile samples matched the OTUs found in the stool samples of the BTC patients, while only 1 out of 11 OTUs (9%) matched in the BBD group (*p* = 0.023 by Fisher’s exact test). We therefore think further investigation is needed; however, it is possible that the combined findings of the common OTUs assigned to *Enterobacteriaceae* and live bacteria isolated from the bile samples are a clue to indicate the relationship of *Enterobacteriaceae* with the onset of BTC. 

Finally, we assessed the genetic features of *Enterobacteriaceae* strains isolated from bile samples in the BTC group. Even though we unfortunately failed to isolate pure strains, our metagenomics data showed the presence of a possible colibactin-producing *E. coli* in a BTC subject. This pathogenic *E. coli* bacterium produces a hybrid peptide-polyketide genotoxin that causes DNA double-strand breaks and the activation of the DNA damage checkpoint pathway, leading to cell cycle arrest. The genetic region responsible for colibactin biosynthesis consists of approximately 20 genes, and its presence has been observed in some *E. coli* strains only [21]. Infection with this colibactin-producing *E. coli* strain has been shown to increase the degree of carcinogenesis of *E. coli*-associated cancer [38]. To the best of our knowledge, we indicated the presence of *E. coli* possibly carrying the *pks* gene island in the bile sample. Whether this bacterium is involved in carcinogenesis in the biliary tract region as well as in colorectal cancer requires further investigation. In patients with BTC, the OTUs of the strains that were significant in the feces were found to be more present in the OTUs of the biliary microbiota, which indicated that the biliary microbiota is influenced by the fecal microbiota. Further studies are needed to reveal whether dysbiosis is the result or cause. However, we would say that the detected colibactin gene cluster might be related to cancer. The results of this study serve as a foundation for further study of the biliary microbiota of the biliary tract.

The limitations of this study are described below. This study was conducted in a single center, and few cases of bile microbiota were evaluated from the bile culture. BTC can be classified into intrahepatic, extrahepatic, and gallbladder types, but due to the small number of epidemiological cases, it is difficult to strictly separate and analyze these types. Therefore, in this study, all three types of BTC were included in the analysis. However, the novelty of this study is that we were able to analyze the same group of background factors, which allowed us to identify the bacterial microbiota-associated characteristics of BTC. Future studies are needed to determine whether the bacteria carrying the *pks* gene are involved in carcinogenesis at the cellular level in BTC. The analysis of the bacterial microbiota of BTC using a larger number of cases from multiple centers is also desirable.

## 5. Conclusions

This study revealed that the bile microbiota of patients with BTC was affected and altered by the fecal microbiota more than that of patients with BBD and that of healthy individuals. This result indicates that the gut microbiota, particularly the colonic microbiota, may affect the bile microbiota of patients with BTC.

## Figures and Tables

**Figure 1 cancers-14-05379-f001:**
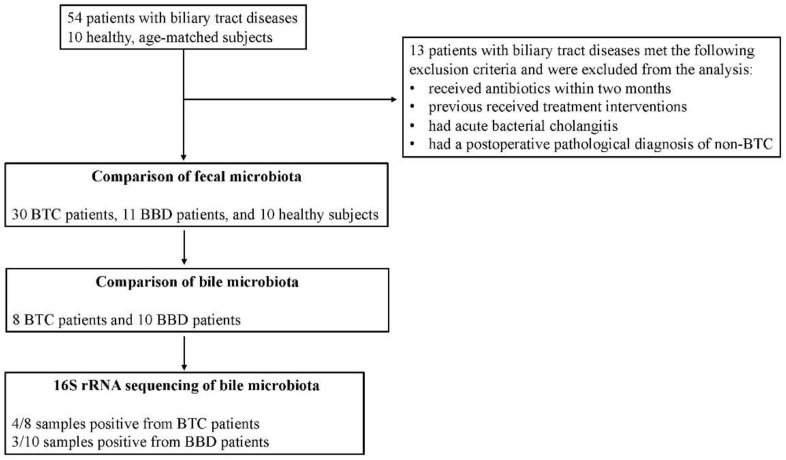
Overview of patient selection and analysis of each sample in this study.

**Figure 2 cancers-14-05379-f002:**
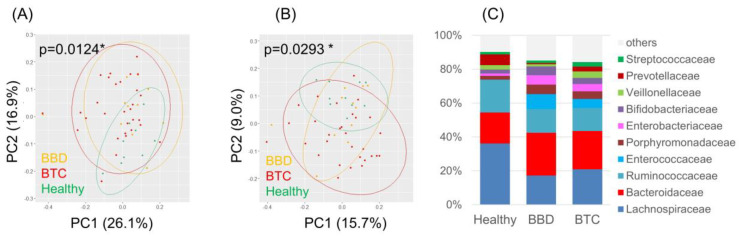
Overview of the fecal microbiota in patients with BTC, patients with BBD and healthy subjects. (**A**) Weighted and (**B**) unweighted UniFrac PCoA of the fecal microbiota; (**C**) Composition of the fecal microbiota at the family level. * *p* < 0.05.

**Figure 3 cancers-14-05379-f003:**
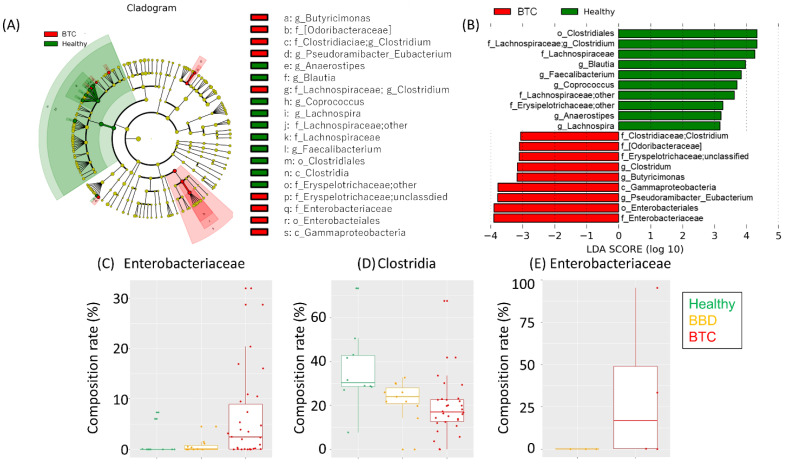
Characteristics of the fecal microbiota in each group. (**A**) Cladogram and (**B**) LDA score based on LEfSe of the fecal microbiota composition of (**C**) *Enterobacteriaceae* and (**D**) Clostridia. (**E**) OTUs assigned to *Enterobacteriaceae* between the bile samples from the BBD and BTC groups. The LDA scores and compositions of *Enterobacteriaceae* are shown.

**Table 1 cancers-14-05379-t001:** Clinical and laboratory data of study subjects.

	BTC (N = 30)	BBD (N = 11)	Healthy Subjects (N = 10)	*p* Value
Age, Median (Min–Max)	75.5 (37–87)	66 (49–80)	63.5 (58–76)	0.054
Female, N (%)	10 (33.3%)	3 (27.2%)	7 (70%)	0.093
BMI, median (min–max)	21.85 (14.2–33.9)	25.3 (19.2–32.9)	21.92 (18.4–28.1)	0.061
Tumor location, N (%)				
Intrahepatic	12 (40%)			
Extrahepatic	12 (40%)			
Gallbladder	6 (20%)			
Stage, N (%) III and IV	16 (53.3%)			
I and II	14 (46.7%)			
Leucocyte (counts/μL)	7303 (3100–22300)	5400 (3700–12200)		0.375
Neutrophil (counts/μL)	5280 (1700–20100)	2900 (1900–8900)		0.203
Hemoglobin (g/dL)	12.45 (6.8–17.7)	14.8 (12.8–18.1)		0.001
Platelet (×10^4^ counts/μL)	21.1 (22.4–54.5)	22.4 (16.3–43.6)		0.768
C-reactive protein (CRP) (mg/dL)	1.30 (0.05–20.85)	0.13 (0.1–1.24)		0.003
Albumin (g/dL)	3.6 (1.9–4.6)	4.1 (3.7–4.7)		0.001
Lactate dehydrogenase (LDH) (IU/L)	202 (113–410)	187 (157–281)		0.164
T-bil (IU/L)	0.95 (0.3–31.5)	0.90 (0.3–1.9)		0.03
HbA1c %	6.0 (4.3–8.5 ND = 1)	5.7 (5.5–6.8 ND = 2)		0.192
Carcinoembryonic antigen (CEA) (ng/mL)	4.3 (1.9–609 ND = 2)	2.6 (1.9–7.5 ND = 2)		0.067
Carbohydrate antigen 19-9 (CA19-9) (U/mL)	63 (0–17799 ND = 2)	16 (0–56 ND = 2)		0.365

median (min–max), ND: no data BTC: biliary tract cancer, BBD: benign biliary disease, BMI: body mass index, CRP: C-reactive protein, LDH: lactate dehydrogenase, CEA: carcinoembryonic antigen.

**Table 2 cancers-14-05379-t002:** OTUs assigned to Enterobacteriaceae in bile and fecal samples.

		Composition Rate (%)													
	Group	BBD						BTC							
	Subject ID	20		23		34		29		37		40		41	
	Bile Collection Methods	ERC		Operation		Operation		Operation		Operation		ERC		ERC	
OTU ID	Taxon	Bile	Feces	Bile	Feces	Bile	Feces	Bile	Feces	Bile	Feces	Bile	Feces	Bile	Feces
OTU_0002	f__Enterobacteriaceae; g__Escherichia; s__coli	45.75			0.53	0.56		6.29	25.91			88.05	19.59		
OTU_0004	f__Enterobacteriaceae													62.71	
OTU_0006	f__Enterobacteriaceae; g__Escherichia; s__coli	21.19						1.12	4.32						
OTU_0007	f__Enterobacteriaceae; g__Citrobacter; s__									5.77	12.37				
OTU_0013	f__Enterobacteriaceae														16.90
OTU_0014	f__Enterobacteriaceae														15.64
OTU_0021	f__Enterobacteriaceae							12.01	0.65						
OTU_0027	f__Enterobacteriaceae							9.52	0.76						
OTU_0034	f__Enterobacteriaceae; g__Escherichia; s__coli											2.41			
OTU_0045	f__Enterobacteriaceae; g__Citrobacter; s__									0.88	2.26				
OTU_0048	f__Enterobacteriaceae; g__Citrobacter; s__									0.83	1.66				
OTU_0050	f__Enterobacteriaceae							4.10							
OTU_0082	f__Enterobacteriaceae; g__Klebsiella; s__					1.57						0.17	0.21		
OTU_0091	f__Enterobacteriaceae							1.96							
OTU_0096	f__Enterobacteriaceae; g__Klebsiella; s__					1.95									
OTU_0097	f__Enterobacteriaceae; g__Klebsiella; s__					1.30	0.61								
OTU_0131	f__Enterobacteriaceae									0.18	0.52				
OTU_0148	f__Enterobacteriaceae									0.16	0.33				
OTU_0209	f__Enterobacteriaceae; g__Citrobacter; s__						0.46								
OTU_0231	f__Enterobacteriaceae							0.37							
OTU_0261	f__Enterobacteriaceae; g__Klebsiella; s__												0.28		
OTU_0323	f__Enterobacteriaceae; g__Citrobacter; s__						0.18								
OTU_0376	f__Enterobacteriaceae; g__Serratia; s__						0.13								
OTU_0405	f__Enterobacteriaceae; g__Citrobacter; s__						0.11								

Highlighting indicates the OTUs detected in both fecal and bile samples.

## Data Availability

All relevant data are within the manuscript and its Supporting Information files.

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
