# Peer review of "Dysbiosis of the Fecal and Biliary Microbiota in Biliary Tract Cancer"

_cancers, 2022, doi:10.3390/cancers14215379_

Round 1

Reviewer 1 Report

I have a concern about how the patients in different groups presented and were entered into the study.  In the BTC group, did the patients present with jaundice?  It does not appear to be the case because their total bilirubin was 0.95.  I do not understand this.  Did they have a stent placed by ERCP or PTC?  If so, it probably would have been better to culture the bile at the time of that procedure instead of surgery.  I have a similar concern about the BBD group.  You need to describe how these patients presented and how they were managed preoperatively.  Another issue is did they receive antibiotics at the time of surgery? If so, what kind?  This may alter the bile and stool cultures.  Finally, how were the healthy subjects chosen and how was the stool collected.  Was the perineum cleansed and with what?  These details are important because they may alter the results.  Was this an approved study and did the patients sign informed consent?

Author Response

Thank you very much for your review.

The manuscript has been improved according to your suggestions, and these revisions are described below.

  1. I have a concern about how the patients in different groups presented and were entered into the study.  In the BTC group, did the patients present with jaundice?  It does not appear to be the case because their total bilirubin was 0.95.  I do not understand this.  Did they have a stent placed by ERCP or PTC  If so, it probably would have been better to culture the bile at the time of that procedure instead of surgery.  

(Response) Thank you very much for your suggestions.

In the BTC group of this study, there were 4 patients whose chief complaint was jaundice. This is because the majority of patients (who came to Jikei University Kashiwa Hospital) were suspected of having biliary tract cancer based on physical examinations and imaging tests. Therefore, 4 patients had jaundice. BTC patients with jaundice underwent ERC before surgery, and bile samples were obtained at the time of initial ERC. We addressed the information in the Materials and Methods section (Line 93-96).

  2. I have a similar concern about the BBD group.  You need to describe how these patients presented and how they were managed preoperatively.  Another issue is did they receive antibiotics at the time of surgery? If so, what kind?  This may alter the bile and stool cultures.  

(Response) Thank you very much for your suggestions.

The BBD group, included patients who visited Jikei University Kashiwa Hospital. The BBD group included patients diagnosed with gallbladder/common bile ductal stones or gallbladder polyps; the BBD group included patients who had not been administered preoperative antibiotics for 2 months. Thus, no preoperative antibiotics were administered. This is because we also believe that antibiotic administration may affect bile and stool cultures. We addressed the information in the Materials and Methods section (Line 101-107).

  1. Finally, how were the healthy subjects chosen and how was the stool collected.  Was the perineum cleansed and with what?  These details are important because they may alter the results.  Was this an approved study and did the patients sign informed consent?

(Response) Thank you very much for your suggestions.

A healthy subject group was included who visited Jikei University Kashiwa Hospital. They had no history of abdominal surgery, severe medical complications, endoscopic retrograde cholangiography (ERC), surgical treatment, or anticancer therapy, taking into account changes in intestinal bacteria. In addition, the healthy subject group must not have received antibiotics more than 2 months prior to participation in the study. This criterion was the same as the BTC and BBD group criteria. Stool samples were collected on the morning of the study day, and the stool sample volume was mixed with 1 ml of guanidine thiocyanate (GuSCN) solution (Techno Suruga Laboratory, Shizuoka, Japan), immediately frozen at -80°C, and stored until analysis. The perineum is not washed. The method of stool collection was the same for the BTC, BBD, and healthy subject groups. This study is of course an approved study, and all patients have signed informed consent forms. We addressed the information in the Materials and Methods section (Line 108-115).

Reviewer 2 Report

Thank you for the opportunity to review this manuscript.

Ito et al. In their study, examine the faecal microbiota of patients with bile duct carcinoma. Furthermore, the connection between the microbial colonization of the bile and the colonization of the intestine is shown. The manuscript is largely clear written. In particular, the methods are described very well and are reproducible. Even if the findings found are by no means uninteresting, I have reservations about publishing the work in its current form.

Major:

1. The study includes only a few patients. In particular, a comparative analysis of the fecal and bilary microbiota only takes place in 4 or 3 individuals. I therefore consider it difficult to transfer the study results to the population.

2. Bile samples were obtained via ERCPs, open and laparoscopic surgery. Here it would be interesting to find out from which subject what type of extraction took place. Possibly as an additional line in the table header of Table 2 or as part of a subject-ID table in the appendix. Was there a connection between the type of bile collection and the success of the amplification?

3. Study participants were recruited at only one hospital. According to the authors, this is seen as an advantage with regard to the homogeneity of the paticipants (genetics, environmental influences, nutrition). In my view, this makes the transferability of the study even more difficult. In their article, the authors name risk factors for bile tract cancer. In view of the number of cases, the risk profiles of the individual participants would be interesting here, which would make the interpretation of the study considerably easier.

4. With regard to the comparability of the groups (BTC, BBD, healthy), which is shown in line 183 and Table 1, I would ask you to reconsider the statistical analysis or the statement made. The lack of statistical significance is not the same as "not different". Using the example of age: A graph of the age of the study participants would show the differences. I would imagine that a comparison between the BTC and BBD groups, without considering the "matched" control group, using a t-test would show a significant difference. However, this assessment can only be conclusively assessed if a list of the study participants with the evaluated parameters is available. If compatible with data protection, I would also like such an anonymous list in the appendix for the reasons mentioned under points 1 - 3.

minor

Fig 1 Please add another box to indicate that the comparative analysis between bile and facies includes only 4 cases out of 8 and 3 cases out of 8 respectively.

Fig 2+3 Please enlarge the axis labeling and the legend significantly

Complete Table 2 Type of bile collection

Author Response

Thank you very much for your review.

The manuscript has been improved according to your suggestions, and these revisions are described below.

Major:

  1. The study includes only a few patients. In particular, a comparative analysis of the fecal and biliary microbiota only takes place in 4 or 3 individuals. I therefore consider it difficult to transfer the study results to the population.

(Response) Thank you very much for your suggestions.

The result of comparative analysis of the fecal and biliary microbiota is reference data due to the small number of cases. Additional experiments cannot be performed in this study. However, we believe that increasing the number of cases and evaluating the results will be a future issue. We addressed the information in the Results section (Line 245-249).

  1. Bile samples were obtained via ERCPs, open and laparoscopic surgery. Here it would be interesting to find out from which subject what type of extraction took place. Possibly as an additional line in the table header of Table 2 or as part of a subject-ID table in the appendix. Was there a connection between the type of bile collection and the success of the amplification?

(Response) Thank you very much for your suggestions.

According to the suggestions, Table 2 has been improved. It was not possible to evaluate which method was more likely to amplify DNA, ERC or surgical collection due to the small number of cases in this study. Moreover, there were no reports that have examined this. We addressed the information in the Discussion section (Line 319-322).

  1. Study participants were recruited at only one hospital. According to the authors, this is seen as an advantage with regard to the homogeneity of the paticipants (genetics, environmental influences, nutrition). In my view, this makes the transferability of the study even more difficult. In their article, the authors name risk factors for bile tract cancer. In view of the number of cases, the risk profiles of the individual participants would be interesting here, which would make the interpretation of the study considerably easier.

(Response) Thank you very much for pointing this out to us. This was an inappropriate statement because the study did not compare the participants to Westerners and did not assess genetics, environmental influences, or nutrition. Therefore, we have removed the statement "may have an advantage with respect to participant homogeneity (genetics, environmental influences, and nutrition)". Diseases that increase the risk of BTC include liver diseases such as cirrhosis, congenital liver fibrosis, metabolic diseases, and primary sclerosing cholangitis (PSC). None of the biliary tract disease patients in this study had these liver diseases. We addressed the information in the Discussion section (Line 90-92, Line 298-301).

  1. With regard to the comparability of the groups (BTC, BBD, healthy), which is shown in line 183 and Table 1, I would ask you to reconsider the statistical analysis or the statement made. The lack of statistical significance is not the same as "not different". Using the example of age: A graph of the age of the study participants would show the differences. I would imagine that a comparison between the BTC and BBD groups, without considering the "matched" control group, using a t-test would show a significant difference. However, this assessment can only be conclusively assessed if a list of the study participants with the evaluated parameters is available. If compatible with data protection, I would also like such an anonymous list in the appendix for the reasons mentioned under points 1 - 3.

(Response) Thank you very much for your suggestions. Since this study compared bacterial microbiota among three groups, we believe that a statistical analysis of three-group comparisons is appropriate. ANOVA was used for the three-group comparison of age. Please do not attach the anonymous list for privacy reasons and because it would be too complicated. We addressed the information in the Discussion section (Line301-304).

minor

  1. Fig 1 Please add another box to indicate that the comparative analysis between bile and faces includes only 4 cases out of 8 and 3 cases out of 10 respectively.

(Response) Thank you very much for your suggestions. Figure 1 has been improved and replaced with new Figure 1.

  1. Fig 2+3 Please enlarge the axis labeling and the legend significantly

Complete Table 2 Type of bile collection

(Response) We apologize for inconvenience. Figures 2, 3, and Table 2 have been improved and prepared in Figures 2, 3, and Table 2, according to the suggestions.

Reviewer 3 Report

1. Did aLl patients undergo ERCP to collect bile samples for the first time? 

2. Please provide the details of benign biliary diseases.

3. Is dysbiosis in bile the cause of BTC or the result of cancer development?

4. Why the alfa-diversity is not different among the three groups?

Author Response

Thank you very much for your review.

The manuscript has been improved according to your suggestions, and these revisions are described below.

  1. Did all patients undergo ERCP to collect bile samples for the first time? 

(Response) Thank you very much for your suggestions. ERC was not performed in all patients. Therefore, some patients in the BBD and BTC groups had bile collected at the time of surgery. We addressed the information in the Materials and Methods section (Line 145-147).

  1. Please provide the details of benign biliary diseases.

(Response) Thank you very much for your suggestions.

Patients treated for gallbladder polyps or gallbladder and common bile duct stones were included. Patients with cholangitis conditions requiring emergency treatment were excluded. Bile was collected from those who gave informed consent prior to treatment; compared with the fecal and bile of BTC, the diseases mentioned above were considered BBD. The BBD group was used as the comparison group because of ethical issues with collecting bile from healthy subjects. We addressed the information in the Materials and Methods section (Line 101-107).

  1. Is dysbiosis in bile the cause of BTC or the result of cancer development?

(Response) Thank you very much for your suggestions. Further studies are needed to reveal whether dysbiosis is the result or cause. However, we would say that the detected colibactin gene cluster might be related to cancer. We addressed the information in the Discussion section (Line 356-358).

  1. Why the alfa-diversity is not different among the three groups?

(Response) Thank you very much for your suggestions. The reason was not sure, but I personally think only certain bacteria (but not a whole microbiota) would induce BTC. We addressed the information in the Discussion section (Line 356-358).

Reviewer 4 Report

Dear Authors,

Thank you very much for your original research article.

I like your article.

I just wanted to ask you to define exactly what BBD without cholangitis is.

Best wishes

Author Response

Thank you very much for your review.

The manuscript has been improved according to your suggestions, and these revisions are described below.

I just wanted to ask you to define exactly what BBD without cholangitis is.

(Response) Thank you very much for your suggestions. Gallbladder polyps or gallbladder/common bile duct stones are often associated with acute cholangitis and cholecystitis. In such cases, antibiotics and endoscopic or surgical procedures may be necessary. Patients who had undergone such prior treatment were not included in this study because of the possibility of further changes in the bacterial microbiota. We addressed the information in the Materials and Methods section (Line 101-107).

Round 2

Reviewer 1 Report

Accept with revisions

Reviewer 2 Report

The authors Zensho Ito et al. succeeded in almost completely dispelling my concerns through the successful revision of the manuscript.

My suggestions were implemented very well or invalidated by adequate and comprehensible arguments. Thank you very much for this.

I recommend publishing the manuscript in the current form.

Thank you for the opportunity to be a reviewer of the manuscript.